# A Lightweight Approach to Localization for Blind and Visually Impaired Travelers

**DOI:** 10.3390/s23052701

**Published:** 2023-03-01

**Authors:** Ryan Crabb, Seyed Ali Cheraghi, James M. Coughlan

**Affiliations:** Rehabilitation Engineering Research Center (RERC) on Blindness and Low Vision, Smith-Kettlewell Eye Research Institute, San Francisco, CA 94115-1813, USA

**Keywords:** blindness, visual impairment, accessibility, navigation, wayfinding, localization, computer vision

## Abstract

Independent wayfinding is a major challenge for blind and visually impaired (BVI) travelers. Although GPS-based localization approaches enable the use of navigation smartphone apps that provide accessible turn-by-turn directions in outdoor settings, such approaches are ineffective in indoor and other GPS-deprived settings. We build on our previous work on a localization algorithm based on computer vision and inertial sensing; the algorithm is lightweight in that it requires only a 2D floor plan of the environment, annotated with the locations of visual landmarks and points of interest, instead of a detailed 3D model (used in many computer vision localization algorithms), and requires no new physical infrastructure (such as Bluetooth beacons). The algorithm can serve as the foundation for a wayfinding app that runs on a smartphone; crucially, the approach is fully accessible because it does not require the user to aim the camera at specific visual targets, which would be problematic for BVI users who may not be able to see these targets. In this work, we improve upon the existing algorithm so as to incorporate recognition of multiple classes of visual landmarks to facilitate effective localization, and demonstrate empirically how localization performance improves as the number of these classes increases, showing the time to correct localization can be decreased by 51–59%. The source code for our algorithm and associated data used for our analyses have been made available in a free repository.

## 1. Introduction

Independent wayfinding is a major challenge for blind and visually impaired (BVI) travelers. The posted signs, landmarks, and other visual cues that sighted travelers take for granted as fundamental navigation cues are partially or completely inaccessible to BVI travelers. Although GPS-based smartphone apps such as Apple Maps, Google Maps, Nearby Explorer (https://tech.aph.org/neandroid/, accessed on 17 February 2023), and BlindSquare (https://www.blindsquare.com/, accessed on 17 February 2023) provide accessible navigation information in outdoor settings, such approaches are ineffective in indoor and other GPS-deprived settings. The limited applicability of accessible wayfinding tools to outdoor settings has serious implications for restricting independent travel by BVI individuals in work, school, healthcare, and recreational settings.

Many localization approaches have been used as the foundation for real-time wayfinding systems. The work of [1] presents a survey of these approaches (see [2] for a recent review of how BVI travelers use wayfinding tools in indoor and outdoor settings). Note that the focus of our paper is on wayfinding rather than the development of mobility aids to warn travelers about nearby hazards such as obstacles and dropoffs [3]. What follows in this section is an updated version of the related work covered in [4].

Past tools for accessible wayfinding relied on special infrastructure such as infrared beacons, RFIDs, and fiducial markers (similar to barcodes). Navigation apps such as NavCog [5] harness Bluetooth low-energy (BLE) beacons, which are another type of special infrastructure used to provide localization information. Unfortunately, such added physical infrastructure has both installation and ongoing maintenance costs, which may limit the adoption of any technologies that rely on it.

Other technologies are now available for supporting indoor localization without the need for any new infrastructure. One such technology that is in widespread use is localization using wi-fi access points [6], which underpins mainstream apps such as Apple Maps (https://www.idownloadblog.com/2019/04/25/apple-maps-indoor-maps-airports-malls, accessed on 17 February 2023) in environments such as airports (https://www.apple.com/ios/feature-availability/#maps-indoor-maps-airports, accessed on 17 February 2023) and shopping malls (https://www.apple.com/ios/feature-availability/#maps-indoor-maps-malls, accessed on 17 February 2023). Magnetic signatures [7,8] are also used as a primary source of localization information in the IndoorAtlas (https://www.indooratlas.com/, accessed on 17 February 2023) and Microsoft Path Guide (https://www.microsoft.com/en-us/research/project/path-guide-plug-play-indoor-navigation/, accessed on 17 February 2023) apps. However, magnetic signatures have significant limitations, including their need for prior calibration, their poor performance in environments lacking extended metallic structures (e.g., wooden buildings), and tendency to drift over time due to environmental changes such as movements of metal shelves and tables.

Another approach is to apply the smartphone’s built-in inertial measurement unit (IMU) to perform step detection and estimate relative movements [9,10], but drift is also a problem unless other forms of location information are used. We note that in future work we hope to augment our localization approach with inertial sensing to lessen its dependence on video imagery, which can be challenging for the user to acquire while walking. While inertial sensing is already integral to the visual-inertial odometry approach we use in our localization algorithm, the use of the IMU to interpret gait movements would allow our algorithm to function even in the absence of video data, e.g., when the smartphone is carried in the pocket or purse.

Another powerful indoor localization approach that eliminates the requirement for additional infrastructure [11,12,13] is computer vision. In some applications, computer vision is used in combination with minimal added infrastructure [14], such as the printed markers used in NaviLens (https://www.navilens.com/en/, accessed on 17 February 2023) and [15]. Although dedicated hardware such as Google Tango is required for some computer vision-based localization systems, other systems such as VizMap [11] use off-the-shelf smartphones. The special localization functionality once available in Google Tango is now implemented in the ARKit and ARCore augmented reality libraries, which run on iOS and Android devices, respectively; in fact, in the Google visual positioning system (VPS) (https://ai.googleblog.com/2019/02/using-global-localization-to-improve.html, accessed on 17 February 2023), localization is performed by using a combination of GPS, Wi-Fi and ARCore. The last-few-meters wayfinding problem was tackled by [16], in which noisy location estimates from GPS, which lack sufficient accuracy to guide a traveler to a precise destination, are augmented by computer vision algorithms for recognizing landmarks, reading signage and identifying locations. The CaBot system [17] is a navigation tool designed to estimate a BVI traveler’s location and provide navigation assistance to a desired destination by using a wheeled autonomous robot the size of a suitcase that automatically avoids obstacles such as people nearby. Finally, deep learning is an important component of our current approach (see Section 2.2.2), and indeed it is an increasingly used tool in localization [18].

Augmented reality libraries such as ARKit and ARCore, which combine computer vision with inertial sensing, are increasingly used to enable localization and navigation functions [19]. The marker-based system of [20] uses the known locations of uniquely identifiable markers on a floor plan to calculate the user’s location when a marker is visible and visual–inertial odometry (VIO, a key function of ARKit; see Section 2.2.1) to determine the location when no marker is visible, based on the relative movements estimated since the last marker detection. By contrast, our approach is markerless and relies instead on visual landmarks such as informational signs and objects already in the building environment such as posters and fire alarm boxes. The Clew [21] app uses VIO to allow a user to “record” a route in one direction and then provides guidance to help the user retrace the route. We note that standard mapping apps such as Google Maps now include augmented reality directions, displayed as arrows and other graphic elements on the smartphone screen, in selected environments (https://waymapnav.com/indoor-navigation-technology-a-comparison/, accessed on 17 February 2023).

This paper builds on past work [4] by using an ARKit-based location approach for indoor environments, with experiments demonstrating its feasibility with BVI participants. The past algorithm was implemented as a standalone smartphone app that ran in real time (approximately 10 frames per second on an iPhone 8). The localization algorithm was the foundation for a second app that provided real-time verbal wayfinding directions (such as “turn left”) to a desired destination [22]. In our past work, the landmark recognition used in the localization algorithm was limited to the recognition of exit signs, and a simplified appearance model was used to provide evidence for specific locations given recognized landmarks. Since then, we have expanded the recognition of visual landmarks to include multiple classes, and the particle scoring has made more effective use of visual information provided by landmark recognition. This paper assesses how localization performance improves with increasing numbers of visual landmark classes.

In past work, we also estimated distances to visual landmarks (at known locations on the map) as a localization cue, using the known height of each landmark above the floor and its apparent elevation above the horizon. Building on our recent work [23], we pursue a more convenient approach by using the heights of the bounding box detections of visual landmarks. This new approach is more convenient because it does not require one to measure the height of each landmark above the floor, only the physical size of the landmark; moreover, distance estimates from the old approach are too noisy to use for landmarks that are at roughly shoulder height, whereas the new approach accommodates landmarks at any height off the ground.

The motivation for our approach is described in detail in Section 4. In brief, our approach is lightweight, requiring only a 2D map of the environment instead of a detailed 3D scan. The approach is also lightweight in that we expect it can run in real time on a smartphone, because an earlier version of the algorithm ran in real time on an older iPhone model, but that is not the emphasis of this paper. It is also privacy-respecting in that only selected visual landmarks need to be photographed to train the localization model, instead of having to acquire imagery of the entire environment. The algorithm is robust to superficial changes (such as new carpeting, moved tables, etc.) that do not affect the landmarks, which means that the localization model does not need to be updated unless there are major structural changes to the environment. However, it is straightforward to update the localization model to add or change the visual landmarks. Finally, the localization algorithm is accessible to BVI users, which means that the algorithm can serve as the foundation for a fully accessible navigation app (see Section 4.3 for details).

## 2. Materials and Methods

This section begins with an overview of our localization approach, followed by details describing the localization algorithm and how data was collected.

### 2.1. Overview of Localization Algorithm

Our localization algorithm combines multiple sources of location information, obtained from a smartphone, over time. After enough time has elapsed to accumulate sufficient evidence, the result is a single location estimate (x,y) that is updated over time. The (x,y) coordinates are defined relative to a 2D floor plan, (which we refer to as the “map”). These sources of location information can be categorized into two types: absolute, meaning information that refers directly to the map coordinates, and relative, meaning information about relative movements (e.g., the user moves in a straight line for 4 m, then turns left by $30^{\circ }$, etc.). Note that the localization problem is easier to solve if we estimate the bearing, denoted by θ, jointly with *x* and *y*. The bearing is the direction defined by the camera’s line of sight projected onto the horizontal plane. It is closely related to the yaw estimated by iOS ARKit, as we will describe later; note that the bearing is a 2D concept, and it is undefined if the camera line of sight is perfectly vertical. We emphasize that our approach is lightweight in that it relies on a 2D floor plan rather than a 3D model of the environment, exemplified by approaches such as ARKit mapping [19] that require the entire environment to be scanned to create a 3D model.

Absolute location sources include the following.

First is the map itself, which specifies the locations of walls, the locations of points of interest such as rooms (specified by name or number), features of interest (e.g., drinking fountain, reception desk), elevators and stairwells, as well as the locations and orientations of signs and other visual landmarks. These locations are specified in a 2D coordinate system that may be converted to GPS or other world coordinates. The impossibility of walking through walls and other barriers is an important constraint that allows the algorithm to rule out many incorrect location hypotheses.Secondly, recognition of informational signs and other visual landmarks using the smartphone’s camera, which constrains the current location relative to the detected landmark(s) on the map. For instance, if a visual landmark that is unique in the building is recognized, then the landmark must be close enough to the current location to be visible to the smartphone camera. The location of the visual landmark in the camera image, and its apparent size in the image, imposes additional constraints on the current location, as we will describe in Section 2.2.2.

There is one main source of relative location information, which is known as visual–inertial odometry (VIO). VIO is a key function [6] supported on modern smartphones that combines computer vision and inertial sensing to track changes in the smartphone’s 6D
*pose*, i.e., 3D location (*X*, *Y* and *Z*) and 3D orientation (pitch, roll, and yaw). VIO is an integral part of simultaneous localization and mapping (SLAM) algorithms, which simultaneously reconstruct the environment in 3D and the camera’s 6D pose on the fly [24]. It is straightforward to relate the pose to the vertical (gravity) direction, which means that VIO data can be directly related to the 2D map coordinates (x,y) and bearing θ.

All sources of location information contain noise and ambiguity that limit the certainty of the location estimates. There may even be inaccuracies in the map itself, which adds additional uncertainty. Thus, if the algorithm is initialized with limited information about the user’s pose (e.g., the user is standing at an unknown location on the map and the bearing is unknown), then a sufficient amount of sensor data will need to be acquired over time before the algorithm can provide a location estimate. To fuse multiple sources of location information and incorporate evidence over time, the algorithm is based on a standard method called Monte Carlo localization [24]. The key to this approach is to represent the knowledge of the current location probabilistically, and to update this knowledge recursively as new sensor information becomes available. This representation and update process is implemented by using a particle filter [24], which we describe in detail in the following section.

### 2.2. Algorithm Details

This section begins with a description of the Bayesian localization model. The subsections that follow describe the implementation of this model.

#### 2.2.1. Bayesian Model

In this model, we refer to the 3D pose $\vec{S}=\left(x,y,\theta \right)$, which is the combined location and bearing that we want to estimate over time. Please note that this 3D pose is different from the full 6D pose associated with VIO. With the pose defined in this way, we can use a hidden Markov model [25] to model the localization process, as is standard in Monte Carlo localization. The model is comprised of a prior model that characterizes the user’s likely motion over time and a likelihood model that describes how VIO and image data relate to the user’s location and motion.

The prior model has two components. The first component is an initial prior $P({\vec{S}}_{t=0})$, which encodes the knowledge at time t=0 about the initial pose. In the worst case (called “global localization”), we know only that the user is somewhere on the map, which equates to a uniform prior jointly over the map region and over the bearing. We will consider other realistic cases below (Section 3.3), in which there is less uncertainty in the initial pose. The second component is a transition model $P({\vec{S}}_{t+1}|{\vec{S}}_{t})$, which characterizes likely movements over the course of a time step, i.e., from time *t* to t+1. Although this component could be used to enforce the fact that the pose typically changes only a small amount from time *t* to t+1 (which typically corresponds to a time interval of roughly 0.1 sec), the evidence from VIO (see below) is strong enough that we do not need to explicitly include this term in the transition prior. Instead, the transition prior enforces a separate, and very powerful, constraint: the fact that the user cannot pass through a wall or other barrier on the floor plan from time *t* to t+1. In other words, any consecutive poses ${\vec{S}}_{t}$ and ${\vec{S}}_{t+1}$, such that the line segment connecting $\left(x_{t},y_{t}\right)$ with $\left(x_{t+1},y_{t+1}\right)$ intersects a wall or other barrier has probability $P({\vec{S}}_{t+1}|{\vec{S}}_{t})=0$. The floor plan is represented as a binary image, with white pixels corresponding to open space and black pixels corresponding to walls and other barriers, so detecting such intersections amounts to a simple ray-tracing calculation. Note that the prior model is conditioned on the specific floor plan of interest, but for simplicity this dependence is omitted in our transition probability notation.

The likelihood model also has two components (see Appendix A for details). The first likelihood component, called the movement likelihood, uses VIO measurements to predict how the pose evolves from time *t* to t+1. The VIO measurements include (a) relative 3D translation X,Y,Z, where (X,Y,Z)=(0,0,0) is the value upon initialization of ARKit, and where the +Y direction is up with respect to gravity; and (b) 3D orientation values of yaw, pitch and roll, in which the yaw ϕ is initialized to 0 upon initialization of ARKit. Note that the 2D VIO measurements (X,Z) are related to the map (world) coordinates (x,y) contained in the pose $\vec{S}$ by an unknown translation and rotation (and note the fact that the *Y* VIO coordinate is unrelated to the *y* map coordinate); similarly, the yaw ϕ is related to the bearing θ by an unknown offset (which drifts only slowly over time). We use the Gravity World Alignment option in ARKit (https://developer.apple.com/documentation/arkit/arconfiguration/worldalignment/gravity, accessed on 17 February 2023), which uses a coordinate system in which *Y* is aligned with gravity and the *X*, *Z* coordinates are arbitrarily aligned to the horizontal plane rather than to the 2D map coordinates (x,y). Thus, changes in ϕ predict changes in the bearing θ. Similarly, changes in the 2D VIO coordinates (X,Z), coupled with the initial bearing $\theta_{0}$, predict changes in the map (world) coordinates (x,y). (In the case that the camera is initially pointed along the map coordinate direction *x* when ARKit is launched, the VIO *X* axis aligns with the map coordinate *x*, but the VIO *Z* axis is antialigned with the map coordinate *y*.) Both sets of predictions must include a small amount of noise to account for drift and noise in VIO measurements.

The second likelihood component, which we call the appearance likelihood, models the detection of visual landmarks (such as informational signs), which have known location and orientation on the map. More specifically, this component models the location in the image a visual landmark is expected to appear, and its estimated distance from the camera, given the pose $\vec{S}$ and the map. The component contains several factors that are multiplicatively combined: (a) to be visible to the camera, the landmark must be positioned in front of (rather than behind) the camera; (b) the orientation of the landmark must also be appropriate to be visible to the camera (e.g., for a one-sided sign, the camera must face the visible side of the sign); (c) the distance from the camera to the landmark, estimated by comparing the known physical dimensions of the landmark with its apparent size in the image, must be consistent with the distance implied by the pose $\vec{S}$ and the landmark’s location on the map; and (d) the measured azimuth, i.e., apparent angular location of the landmark in the image along the horizon line (left, right, or center), must be consistent with its azimuth implied by the pose $\vec{S}$ and the landmark’s location on the map.

Next, we address two specific issues that apply to the likelihood component: (a) the possibility of multiple landmark detections in a single video frame, and (b) the unknown correspondence between physical landmarks and detected ones. Regarding issue (a), if there is more than one detected landmark in an image, then the overall appearance likelihood is calculated by evaluating the appearance likelihood separately for each detection and multiplying these likelihoods together. Issue (b) relates to the fact that the identical visual landmark (such as an exit sign) may appear in multiple places on the map, so when the landmark is detected, it is unclear to which physical landmark it corresponds. We solve this by using a greedy procedure for each detected landmark, in which we search over all corresponding physical locations and choose the one resulting in the highest appearance likelihood for that detection. Note that we choose to recognize classes of landmarks that are sufficiently visually distinct so as to minimize the incidence of misclassifications (confusing one class with another).

The overall likelihood is the product of the movement likelihood and the appearance likelihood, where the movement likelihood equals 0 if the path has intersected a wall and equals 1 otherwise, and the appearance likelihood itself equals 1 in a frame with no detections and equals the product of appearance likelihoods for one or more detections in the frame.

#### 2.2.2. Visual Landmark Recognition

Building on the work in [4] that relies on exit signs as visual landmarks using an AdaBoost Exit sign detector, we expanded our framework to flexibly support many classes of visual landmarks by using an off-the-shelf deep convolutional neural network object-detection algorithm that can be easily trained for multiple simultaneous classes. Keeping in the same vein as posted exit signs, we generally used flat (2D) informational signs as the visual landmark classes, but explored some variations including items such as artwork, nonplanar objects such as a fire alarm or fruit basket, and large posters (see Figure 1 for some examples and Appendix B for a complete listing of classes).

Landmark classes were generally chosen based on a few criteria: visibility, location on the map, detectability by an algorithm (i.e., having distinct visual features and high contrast), and being sufficiently distinct from other classes. We created a separate map for each floor of our building, and created a separate object recognition model for each one. That is, for each map we trained an object detector to recognize a set of landmark classes specific to that environment, with a slightly different focus of landmark types on each floor. In the garage environment, for example, we included classes for parking space numbers painted on the floor and wall; although the overall appearance of each instance was similar (e.g., a bright yellow number on a red square) there were distinct differences from one to the next as each has a different number, and there were many instances of these classes spaced regularly throughout the environment. On one floor, we posted seven unique letter-size sheets of colorful art prints to the walls, whereas on another floor we used only landmarks that were already present. There were four classes, including exit signs and fire alarms, that were included in each space. See Figure 1a for examples of the landmarks we trained.

We specify that the output from an object-detection algorithm should include the class of a detected object, its size and location in the image (whether from the bounding box or labeled pixels), and a confidence estimate of that classification. In this work, we use the You Only Look Once (YOLOv2; [26]) object-detection model, and describe the training and usage of the system in Section 2.2.7 about implementation. The motivation for choosing YOLO as a replacement for the Adaboost cascade used for object recognition in an earlier version of our localization algorithm [4] is that YOLO has recognition accuracy that is superior to that of older approaches, such as the Adaboost cascade, it can seamlessly recognize dozens of class simultaneously, and it performs fast enough to run at frame rate on a mobile device such as a smartphone (see Section 2.2.7).

Given the physical dimensions of a visual landmark (assumed to be planar or nearly so), we use the height of the bounding box (see the yellow rectangles in Figure 1b) to estimate the distance to the landmark. First, the image is unrolled so that it is upright with respect to gravity, performing a 2D image rotation by the amount specified by the ARKit roll (one of the three Euler orientation angles estimated by ARKit (https://developer.apple.com/documentation/arkit/arcamera/2866109-eulerangles, accessed on 17 February 2023), in addition to pitch and yaw). Note that the black pixels visible in the borders of the image in Figure 1b are created in the unrolling process due to filling in the pixels that are not visible in the original camera frame. Unrolling the image generally makes the bounding box tightly frame the landmark, (which typically has a rectangular border that is aligned to gravity), because the border is more likely to appear horizontal in the unrolled image. Following the work of [23], we estimate the distance to the sign (more accurately, the distance of the line connecting the camera to the sign centroid projected onto the ground plane) as a function of the apparent height of the landmark’s bounding box (the difference between the bottom and top pixel row coordinates of the landmark’s bounding box in the image); note that the apparent height increases as the landmark approaches the camera. Because of fluctuations in the bounding box size returned by YOLO (discussed in Section 3.6), and because of situations in which the bounding box is poorly aligned to the borders of the landmark (e.g., the yellow “5” sign shown in the bottom of Figure 1b), the distance estimate is noisy. As a result, we have chosen the distance estimate to be weighted only lightly in the model.

For one class of landmarks, parking space numbers painted on the ground, we used the same approach as in [22] to estimate the distance to the landmark. In this case, we used the downward angle toward the landmark’s centroid with respect to the horizon and an estimate of the camera height to make this determination. A single approximate estimate of 1.64 m was used for all participants, though future work could include a method by which to calibrate this value for individuals.

We can determine the direction of the landmark, i.e., the azimuth angle with respect to the device, by using the centroid of the bounding box in the unrolled image (specifically the pixel column of the centroid) in conjunction with knowledge of the camera’s intrinsic geometry. The azimuth estimate tended to be less noisy than the distance estimate, so we weight the azimuth cue more heavily than the distance cue in the model.

#### 2.2.3. Particle Filter

The Bayesian localization model is implemented by using a particle filter [24] (see [27] for a tutorial on particle filters). In the particle filter framework, we track and evaluate a large set of so-called particles, whereby each particle may be thought of as a hypothesis of the current pose $\vec{S}=\left(x,y,\theta \right)$. The entire set of particles represents a set of random samples from the current posterior distribution of the user’s pose. The set of particles is initialized by sampling from a distribution informed by prior knowledge and assumptions of the situation (e.g., uniform over the space of the map coordinates and bearing in the case in which we assume only knowledge of which floor the user is on). With each time step, a new VIO reading and image are obtained, and the particles are updated by using this new information. The translational movement from the previous VIO pose to the current pose is applied to each particle’s position, after first being rotated to account for the difference between the device’s yaw (as reported by VIO) and the particle’s hypothesized bearing. We add some additional translational noise, the variance of which is proportional to distance moved, and some bearing noise, of a constant variance, to account for measurement noise and other sources of uncertainty (including inaccuracies in map scale or landmark positions, drift in ARKit’s coordinate system, and positional errors in object-detection results).

After particles are updated, the hypothesis likelihoods are reevaluated for the resampling stage of the particle filter. First, however, any particle whose path has intersected a wall or boundary (as defined by the map) are immediately removed. To avoid issues caused by too frequent resampling, we only resample the particles when fewer than 10% of the hypotheses remain (due to hitting a wall) or after a detection event, when one or more of the object classes on the floor has been detected. If no landmarks are detected on the current frame, each particle is assigned equal likelihood; otherwise the likelihood score for each particle is determined based on the object detection as described in Appendix A. We then sample from the set of particles, with replacement, in proportion to their likelihood score until we have a full set of *N* sampled particles ready for the next frame.

To get useful output from our distribution of particles, we need to extract a localization estimate from the set of hypotheses comprising the particle filter. To accomplish this, we employ an approximation of a kernel density estimator (KDE) to calculate the probability density function (PDF) of the pose $\vec{S}$, discretized over (x,y) locations and over the bearing θ. Once the PDF has been calculated, we determine the locations of peaks in the PDF. Specifically, we use the pixel grid of the map for position binning and divide the bearing space into four quadrants, creating an W×H×4 volume. The binning over bearing space has two chief implications: first, hypotheses that are markedly different in their bearing even if they reside in the same area on the floor plan do not support each other (in terms of evidence that the correct location is in that area), and, in addition, this means that in the case of constrained localization where we assume the approximate start location (detailed in Section 3.3), the algorithm will not start reporting the user’s location until their bearing has been disambiguated.

Iterating over the particles, we add the likelihood score of each to its associated voxel and then approximate a Gaussian KDE by performing a Gaussian blur on each of the slices in the *x*-*y* plane. Each peak (defined as being no less than any of its eight nearest neighbors and greater than at least one of them) is considered a candidate for the user’s location. In the context of guided navigation, we consider it to be important to avoid giving incorrect directions, so we prefer to reach a certain level of confidence before reporting a location; thus it is not until the highest scoring peak is at least twice the score of the next highest that the algorithm reports a location, which we refer to as localization convergence. As part of a navigation app as described in [22], the user’s estimated pose is used to provide real-time navigation guidance.

The particle filter is initialized in a way that depends on what information is known about the user’s starting position. For the global localization condition (described in [24], the starting position and bearing are unknown, except that the camera is known to be on a specific map, i.e., a specific floor), particles are randomly generated such that the locations are uniformly sampled across the area represented by the map and the bearings are uniformly sampled across all possible directions (from 0 to 2π) as illustrated in Figure 2a. In another initialization condition, described in Section 3.3 and referred to as constrained localization, the location is assumed to lie in a circle of radius *R*, and the bearing is unknown. In this case, particles are randomly generated with locations uniformly sampled across the area of the circle and the bearings are uniformly sampled across all possible directions.

The algorithm, as described here, is summarized in Appendix A.

#### 2.2.4. Unique and Nonunique Landmarks

Some visual landmarks are unique in a setting, such as a special art poster that appears in just one location on a floor plan. Other landmarks, such as exit signs, appear in multiple places on a floor plan, often with identical appearance at each location. In fact, in some instances both sides of an exit sign appear identical. We refer to this case as a nonunique landmark. Whereas recognition of a unique landmark implies that the camera is somewhere near the landmark, recognition of a nonunique landmark poses a fundamental ambiguity: all we can conclude from a single image with this recognition is that the camera is somewhere near one of the instances of the landmark on the floor plan.

Fortunately, the Monte Carlo localization approach we use (Section 2.2.1), implemented by a particle filter, can accommodate evidence from both unique and nonunique landmarks. Recognition of a nonunique landmark means that all particles sufficiently near every instance of the landmark will receive high scores. This is a source of uncertainty that sometimes manifests as multiple clumps of high-scoring particles in the particle filter, instead of a single clump that might result from recognition of a unique landmark. The accumulation of evidence from multiple video frames over time allows the localization algorithm to recover from the ambiguity, resulting in a single location estimate.

We note that there is a tradeoff in the usefulness of unique and nonunique landmarks. On the one hand, there is no ambiguity when recognizing a unique landmark, which makes the evidence from this recognition definitive. On the other hand, if there are few classes of landmarks, then there are few opportunities to encounter and recognize a unique landmark, whereas there may be many opportunities to recognize nonunique landmarks, such as exit signs, throughout a floor.

Section A.1 formalizes the way that nonunique landmarks are handled by the localization algorithm. This is accomplished by casting the ambiguity as an unknown correspondence problem.

#### 2.2.5. Example of Particle Filter in Action

This subsection explains Figure 2, which demonstrates how the particle filter updates the pose distribution over time. The left side of the figure shows a portion of the map, with landmarks drawn in color (green represents exit signs, red is for fire alarm, and yellow for caution signs). Each landmark is drawn as a either a half ellipse or full ellipse, with the curved portion of the ellipse representing the visible side of the landmark (because each landmark is either planar or nearly so). Some exit signs have signs visible on both sides, in which case the landmark is drawn as a full ellipse along with the major axis indicating the plane of the sign. The location of each particle is represented by a gray point, with a subset of the particles (to avoid overcrowding the image) represented by arrows, pointing in the bearing direction with the length of the arrow proportional to the likelihood score of that hypothesis.

On the right side of the figure is the image captured at five moments in time (subsampled from data collected at roughly 20 frames per second, as described in Section 2.2.7), with landmark detection results annotated with a colored bounding box. Note that the landmark recognition confidence threshold is set to 70%, so only detections with at least that confidence are considered. We comment on each of the five moments included in the figure. First, we have (a) (t = 0 s). This is maximum uncertainty after being initialized with a uniform distribution: there are no landmark detections, and we draw the particles as triangles to indicate their bearing. Secondly, we have (b) (t = 0.05 s). One detection has crossed the threshold (exit), and we can see how the particles pointing toward exit signs on the map are longer than other particles, because the location and bearing of these particles is consistent with recognition of an exit sign in the image. Thirdly, we have (c) (t = 0.4 s). After some time, we can see particles whose bearing faces the signs at the appropriate distance have started converging on the exit sign locations. Still, there is ambiguity as to which exit sign on the map has been detected and we see some clustering in multiple locations. Fourth, we have (d) (t = 1.15 s). The detection of another landmark (fire alarm) starts to disambiguate the uncertainty in (c). Finally, we have (e) (t = 3.25 s). All of the particles have converged to a single cluster (where the particles have nearly identical bearing), and the localization result will be reported successfully.

#### 2.2.6. Mathematical Foundations

This subsection succinctly summarizes the mathematical foundations of our approach. Overall, the localization model is based on a Bayesian approach called Monte Carlo localization [24], which is a hidden Markov model [25] that represents the probabilistic knowledge of the current 2D location (x,y) and bearing θ given all available data at each time step: camera and sensor data in the form of VIO and landmark detections, and the impassability constraints implied by the 2D floor plan. VIO data provides strong evidence for relative movements through the environment, but on its own it does not provide information in the absolute reference frame of the floor plan (e.g., bearing pointing north, or a location near a specific office). This information about relative motion is augmented by absolute information from landmark detections (provided by YOLO object recognition), which give strong evidence about the current location and bearing, and from the placement of walls and other structural barriers in the floor plan that the estimated trajectory can’t intersect. Perhaps the most elaborate part of the probabilistic model is the likelihood model, which uses 2D geometry to enforce constraints that landmark detections impose on the possible location and bearing (see Appendix A).

This probabilistic model is implemented by using a particle filter [24], which can be regarded as a tool that approximates the probability distribution over the pose (location and bearing) to be estimated at each time step. The probability distribution is approximated by a large number of particles, each of which specifies a specific pose hypothesis. Regions of pose space with a higher density of particles correspond to higher probability estimates, whereas regions of pose space with fewer particles correspond to lower probability estimates. When the particles are clumped sufficiently tightly in a cluster that is bigger and tighter than any other clusters, we can form an estimate of the current location and bearing from this cluster.

The particle filter is initialized according to whatever knowledge is available when the algorithm is begun: for global localization (see Section 3.1), the particles are initialized randomly and uniformly across the entire map, whereas for constrained localization (see Section 3.3), the particles are initialized randomly and uniformly across a small circular region centered on a specific location on the map. After some time has elapsed, the algorithm converges to an estimate of the current location and bearing, which is updated in subsequent time steps of the algorithm.

#### 2.2.7. Implementation

Our study is based on real-time data collected by an iPhone app that we previously developed [4], called CameraLog, which acquires and logs timestamped video and ARKit data for offline analysis. The localization algorithm itself (including YOLO) was implemented and run in Python. The particle filter used *N* = 100,000 particles.

To train YOLO, we used the Turi Create package (https://github.com/apple/turicreate, accessed on 17 February 2023), and we used the Core ML Tools package (https://coremltools.readme.io/docs, accessed on 17 February 2023) for object recognition. Both are released by Apple, and the Core ML models that Turi Create produces can be used directly by the iOS Core ML software. An object-detection model was trained for four environments (one for each floor of the building), with 7–11 classes for each model, respectively, and training data included labeled bounding boxes for roughly nine examples (median) per class (around 40 for exit signs and as many as 64 and 110 for parking space numbers). Photos were captured by using an iPhone 11 Pro of landmarks in situ, often with multiple landmarks in each image (on average 1.7 landmarks per image), and for classes with multiple instances (e.g., exit signs) we included multiple examples, but not exhaustively. Models were trained over approximately 20,000 epochs, requiring approximately eight hours on an Apple M1 processor. The trained models perform detection on a 416 × 416 pixel scaled image, which takes approximately 21 ms on an Intel Iris Plus Graphics 655 GPU.

We note that in our past work [4] we implemented a very similar localization algorithm in Swift and C++ as an app running on an iPhone 8 by using *N* = 50,000 particles. The main difference with respect to the current algorithm is that the old implementation used an AdaBoost-based exit sign-recognition approach, whereas the new localization algorithm uses YOLO to recognize multiple classes of visual landmarks. The current implementation, written in Python (and not particularly optimized) runs at approximately 2–3 frames per second on a 2.7 GHz Quad-Core Intel Core i7. We plan to port the new localization algorithm to the smartphone platform in future work. Given that the Swift/C++ version of the localization algorithm ran at 10 frames per second on an iPhone 8, we are confident that the new localization algorithm will run at a comparable (or better) frame rate on a newer iPhone, such as an iPhone 14.

Our Python source code, along with maps, trained Core ML models, ARKit’s VIO data, and accompanying images have been made publicly available (https://doi.org/10.3886/E183714V1, accessed on 17 February 2023).

### 2.3. Experimental Methods and Data Collection

In order to test our algorithm under various conditions, we collected data with participants to simulate navigation in an indoor environment, and performed analysis offline. The participants were asked to hold a smartphone (iPhone 11 Pro) as they would if using a camera-based navigation app (upright with the rear-facing camera pointed in front of them); if the camera is tilted too far down or up (more than 20° in either direction) the app warns the user to right it. They were then instructed to walk along a designated route. The routes were designed to return to the starting point and thus form a loop—this way, in offline testing, we generate virtual trials to increase the amount of data we analyzed. Specifically, we can choose to start from any point in the trial and create a loop (closed circuit) by using the remaining points in the trial followed by the earlier points in the trial. Data was captured at a rate of 8.6 frames per second for the first participant, and at about 19.5 fps for the subsequent participants.

Data was collected with six participants: one blind and five sighted individuals, two of whom were female. The data collection procedure was covered by an IRB protocol, which included a consent form for all participants to fill out for participating in the study. Participants were told to walk along predefined routes of the building while holding the iPhone 11 Pro forward (so that the camera line of sight remains roughly horizontal). The blind participant opted to hold the elbow of the safety monitor (a member of the experimental team focusing on maintaining the participant’s safety) whereas in the open environment (although this was not an independent navigation condition, the blind participant aimed the camera independently, without guidance from the experimental team), and in the corridors walked with a white cane.

Routes were on each of the four floors of a single building measuring 20 × 42 m, and each route was a closed circuit; Figure 3b shows an example route on each floor. The routes on the first floor were conducted in a parking garage where the open floor environment comprised the majority of that map, whereas the routes on the other floors were in hallway corridors. Routes did not stray into individual rooms and were contained entirely in the corridors (and in the open space for the garage floor), even though we populated the entire map with particles. There were a total of 37 trials analyzed, averaging 83 m each and covering about 3100 m in total. The participants were asked to begin each route by first panning the smartphone’s camera from one side to the other before walking, as might be used in practice, to allow the algorithm to detect visual landmarks. This instruction was not enforced and participants included the panning motion about half the time. In each trial, the participant began at a specified location, aiming the camera in a specified initial direction (bearing), then walked along the route and was told to stop upon reaching the initial location again. At the very end of the trial, participants were told to change their bearing to match the initial bearing, and thus close the loop.

Having fully completed loops allows us to select any point on a trial’s route and use that as a starting point of a virtual route, knowing that we have continuous motion data from VIO that will be consistent with the users’ motion, even through where we “splice” the end to the beginning of the trial. This will create a jump in the user’s pose at that frame transition, which will appear unnatural compared to other time steps (the size of the jump depends on how closely the participant returned to their starting pose at the end of the trial, and we selected the starting and ending frames so as to minimize the offset), but the relative movement of the camera will be consistent with the ground truth location on the map. This could be disruptive to a motion model (if one were modeling velocity in the particles) or be flagged as possibly erroneous output from ARKit (if we were performing such monitoring, as discussed in Section 4.2). We otherwise see this as a reasonable if not somewhat novel variation of (test–time) data augmentation [28]. For each of our trials, we selected four starting points evenly spaced (by time), starting at the original first frame.

#### 2.3.1. Ground Truth Determination and Uncertainty

To evaluate the performance of the localization algorithm, we determined the ground truth location of the device for each frame recorded. As the participants collected data during trials, they were followed and video recorded at a distance so that afterward the sources were frame-synced and their position on the route could be observed in the accompanying video and marked on the digital map at key frames (see the red dots in Figure 4). The positions between the key frames were estimated by interpolating the relative movements provided by VIO. The veracity of the VIO movements was confirmed by individually inspecting the reported VIO trajectories; in the two cases where the VIO data was found to be unreliable, we increased the density of marked key frames and used simple linear interpolation without incorporating the erroneous VIO movements. We conservatively claim that our ground truth locations were within 1 m of the actual position for every frame.

There are other factors which limit the precision of the system as a whole. There will be inaccuracies of wall placement and of map scale to some degree given any floor plan. We measured the wall-to-wall distance of some of our main corridors with a laser measure, and found the scale for those corridors on the map consistent within 2%. The magnitude of this error will be influenced by the construction of a floor plan for different environments. Similarly, the placement of the visual landmarks on the digital map will have some amount of error, whether the location of each landmark is carefully measured to determine its pixel position, or rather the position is eyeballed and marked on a GUI for the floor plan (as might be likely in a wide-deployment scenario).

Although ideally we would like the ability to evaluate our localization results with centimeter accuracy, the expense and effort required to deliver that level of ground truth accuracy would be considerable. In the context of providing guidance to blind travelers, we don’t specifically have a need for such localization accuracy. We are more concerned with the incidence of catastrophic errors—highly consequential errors that could send the traveler down the wrong corridor—than with the incidence of small ones, and our performance assessment reflects this priority. Here we consider a “small” localization error to be on the order of 1 m, because the traveler can use their cane or guide dog to recover from an error of this magnitude in locating a door or corridor that they wish to arrive at in typical navigation tasks. We also estimate that our ground truth location data has an accuracy of about 1 m, as explained in Section 2.3.1.

## 3. Results

We characterize the performance of the localization algorithm in terms of (a) the distance that must be walked to attain an accurate location estimate and (b) the distance of the predicted position from the ground truth location, the localization error. We refer to the distance walked to attain an accurate location estimate under this condition as the distance to correct convergence, and we calculate the distance traveled by accumulating over the relative motion reported by ARKit, sampling every 20th frame (every 1–2 s). We also report the time to correct convergence, which we found to be highly correlated with the convergence distance, unsurprisingly. The localization error is perhaps the more common metric for evaluating the performance of a localization algorithm, but note that our algorithm suppresses any localization estimate until certain conditions are met, so we can only consider frames for which an estimate of the position is given by the algorithm and omit any frame in which the location is indeterminate. With this in mind, we also look at the overall rate at which trials eventually converge on the correct location.

This performance is evaluated as a function of the number of visual landmark classes that are recognized, from no landmarks, to a single landmark class (exit signs, as in our previous work [4]), to multiple (7–11) landmark classes (which we refer to as “all” classes). Figure 3 row a shows the locations of landmarks in the set of maps we tested. Performance is compared under two sets of conditions: environment type (one dominated by corridors vs. one dominated by wide-open space) and starting condition (global localization or constrained localization). Our main result is that recognizing more classes of landmarks generally leads to a lower distance to correct convergence, and that localization is more challenging in wide-open spaces than in environments dominated by corridors.

### 3.1. Performance of Global Localization Algorithm

In our chief analysis we assume an “unknown starting location” condition, i.e., the algorithm is initialized knowing only which floor of the building map that the user is on. This condition corresponds to the global localization problem as described in [24]. In order to quantify the distance to correct convergence (in shorthand, convergence distance) we must first define what is considered “correct” and choose a threshold at which to consider the localization estimate to be close enough. For the purposes providing turn-by-turn navigation instructions, one could be reasonably successful with localization accuracy within a few meters, though we decided to choose a more conservative value of 1 m, and in the evaluation of our results we refer to localization within this threshold distance as the “correct location.”

First examining the performance in the corridor environment in Table 1, we find that for trials in which the algorithm successfully localizes, the distance and time to correct convergence when using object detection over the full collection of landmarks is approximately one third of that when we employ the particle filter without landmark detection. We observed that using exit signs as the only class of visual landmark still resulted in convergence distances and times of about two thirds that without any detection. Looking at the median localization error, there is no substantial difference whether or not we are using landmark detection in the corridor environment. In terms of the rate of successfully converging on the correct localization, we actually see a slight advantage when suppressing the detection in the corridor environment.

Next, in the open environment of a parking garage we find the advantage of using all of the signs to be even more pronounced for the distance and time to correct convergence over the cases of no landmark detection. Using only exit signs, which are especially sparse on this floor (see Figure 3a-1), the distance to correct convergence was only somewhat shorter than without detection, and remained about the same for convergence time. In this open environment, however, we did observe that the median localization error was significantly reduced by employing landmark detection: about one third and one half, respectively, for all signs and exit signs compared with no detection. Perhaps the most notable result in this setting is that of the overall correct convergence rate; without the aid of visual landmarks, the particle filter converges on a correct location in only 17.5% of the trials compared to 90% rate when using all classes of landmarks.

In Figure 5 we plot the empirical cumulative distribution function (ECDF) of the localization error aggregated over multiple samples of this error, explained in the next subsection.

### 3.2. Empirical Cumulative Distribution Function

The Empirical Cumulative Distribution Function function [29] is the empirical version of a cumulative distribution function, which is a plot of cumulative probability on the vertical axis as a function of the value of the variable of interest on the horizontal axis; given a value *v* on the horizontal axis, the corresponding cumulative probability represents the fraction of data samples whose value is less than or equal to *v*. The value of *v* for which the cumulative probability equals 0.5 is the median of the samples, and where the cumulative probability equals 0 and 1 are, respectively, the minimum and maximum observed values. An advantage of the ECDF over more common visualization tools such as the histogram is that the appearance of the histogram depends heavily on the placement of histogram bins (with wider bins creating a smoother histogram than one with narrower bins), which is arbitrary, whereas the ECDF requires no such choice.

Next, we address the issue of missing data. As described above, in some video frames there is no localization estimate provided by the algorithm, and in some trials there is no distance to correct convergence as the algorithm fails to converge. Rather than simply omitting this missing data from the ECDF plots (which would be misleading, because larger amounts of missing data typically correspond to worse performance of the localization algorithm), we choose to represent each piece of missing data by a distance that is larger than any physically realizable distance (given the size of the building). We refer to this distance as “Inf”; labeling such frames with a value that is larger than any physically realizable distance allows the ECDF plot to indicate the fraction of trials with missing data (i.e., no localization estimate available). The rationale for this approach is that missing location estimates could be replaced by a default location estimate, such as the centroid of the map, and the resulting localization error would be guaranteed to be no worse than the Inf value. Similarly, trials in which no correct convergence has been achieved are counted as missing data trials with an Inf value in the ECDF plots.

In Figure 5 we present the ECDF of the localization error for both the corridor and open environments. We note the dog leg shape of the plots, especially prominent in Figure 5a, and reflect on what underlies this shape in the ECDF of localization error. We found that our localization algorithm tends to exist in one of three states at any given time: a location is reported that is essentially correct (the initial steep portion of the curve where the localization error is expected to be small), an incorrect location is reported (the flattened portion of the curve where the localization error can take on any value within the size of the floor plan), and no location is reported as the confidence is below threshold (the gap from where the line hits the right side of the plot up to 1.0). Ideal performance would appear in this plot as a sharp turn in the top left corner. It is worth noting that when we do not employ landmark detection (denoted as None in the legend) the curve tends to be more flat than the other cases, once it plateaus (and in fact clearly crosses the curve for exit signs only) which seems to indicate that although it is less likely to converge on the correct location, there tend to be fewer occasions when an incorrect location is reported.

In order for our localization algorithm to serve as the foundation for an effective wayfinding aid, we want to begin delivering navigation instructions as quickly as possible, and minimize the amount of unguided exploration we impose on the BVI traveler. Key metrics by which to gauge performance in this regard are the distance traveled before successful localization and the time elapsed. Inspecting the plots of Figure 5 and noting where the dog leg bends (at approximately 1 m), we feel this data supports our choice of what we consider successful localization, being within 1 m of the ground truth position. For the distance before convergence, we plot the ECDF of the distance traveled for each trial before the algorithm locks in to their position in Figure 6a,b.

### 3.3. Constrained Localization

Another set of tests involved the initial condition of known approximate location and unknown bearing. This condition of knowing the approximate location might be encountered in a situation that uses Bluetooth low energy (BLE) beacons (1–5 m) or WiFi access points (10 m), or if a user knows their approximate location (e.g., “the north entrance”) but is unable to pinpoint it on a map GUI. In these cases, we initialized the algorithm with hypothesized positions within a set radius (tested with 1 and 3 m) around the ground truth position of the user at the first frame; bearing is unknown and was sampled randomly and uniformly from 0 to 2π.

Constrained localization is an easier case than global localization, and Table 2 shows that under all conditions every metric performs at least as well as the global localization condition. Looking at the convergence rate, for example, we find that in nearly all cases the algorithm eventually converges on a correct location within that trial, even in the open environment. The localization error, on the other hand, is not substantially improved by the additional information as to where the user starts—in the corridor environment, the median localization error is nearly identical to that of the global localization case, and the improvement in the open environment when using landmark detection could be described as marginal. The only substantial improvement for localization error seems to be in the open environment without the use of visual landmarks.

What we observed for the time and distance to correct convergence is more interesting. Given information about the starting location of the user, the algorithm converges to a correct pose much more quickly whether or not landmark detection is employed, but under this condition we find the advantage in using landmark detection is much more pronounced. Looking at the distance to correct convergence in the corridor environment, if we assume the starting location is known within 1 m then we find without landmark detection it would only take an average of 9.5 m of exploring before the particle filter converges on the correct location (instead of 26 m without initial location information), yet when we include landmark detection it then takes only about 1.6 m of unguided exploration before the algorithm confidently identifies the user’s bearing and location and could begin providing guided navigation. Table 2 only shows the median value (50th percentile), but in Figure 6c,e we can see that the advantage of using visual landmarks is consistent up through approximately the 90th percentile.

Even more interesting is that in the open environment, where even given the approximate starting location, without the aid of visual landmarks the algorithm still takes a prohibitively long time to converge on the correct location (though it does so eventually most of the time here) of about 45 m and takes well over a minute. When using visual landmarks in constrained localization, we find the algorithm’s best performance is in the open environment, with a median time to correct convergence of 1.1 s and 0.4 m. Notably, convergence occurs sooner in the open environment than in the corridors (this is the only condition in which we find better performance in the open environment). Figure 6 also seems to indicate that our algorithm generally performs best in an open environment with the hint of initial location. We explain this apparent anomaly as such: while walls provide a constraint to the allowed movement of particles that tends to accelerate convergence, their absence allows the camera to see many more of the landmarks from any position, and given the approximate location the detected landmarks can help easily determine the unknown bearing. In the corridor environment, we observe correct convergence about 4–6 times faster with all classes of landmarks than without, and in the open environment it is at least 15–100 times faster. By using only the exit signs, which are quite sparse in the garage that was used for the open environment, we observe a more modest advantage of using landmark detection, with correct convergences occurring approximately twice as quickly as without.

### 3.4. Consistency of Performance: Sighted vs. Blind Participants

During our period of data collection for our study, we unfortunately had limited opportunity to work with blind and visually impaired participants. As the target population for our proposed application are BVI, there is potentially some concern that the collected data may not be representative. Here, we briefly provide our reasoning for the validity of our dataset. As a start, we point to our previous work in [4], a study which involved six blind participants who either collected data for offline evaluation (like in this work) or used a live navigation app that used an earlier version of this algorithm for the localization component of the app. This earlier work demonstrates the accessibility of our basic algorithm, and our novel innovations expanding upon this work (to include a larger variety of visual landmarks, the formulation of our appearance likelihood, and use of deep neural networks for landmark detection) do not obviously invalidate this finding.

Still, we were interested in a side-by-side comparison of the data and algorithm performance between our blind and sighted participants. In Figure 7 we use a box plot to visualize the distribution of distance to correct convergence for each of our participants (BP or SP are used to indicate blind or sighted participants, respectively). The distance to convergence for our blind participant may be on the lower end, but the overall distribution of the sighted users is quite in line with the blind participant.

We also looked at the ECDF of the localization error for each of the participants, for the case of using all landmarks. In Figure 8 we see a consistent shape of the curve for all participants, though the location of the dogleg bend varies from about 60% to 80%.

One incidental difference is the frame rate of data captured by the blind participant was lower than for the sighted ones (owing to a difference in how the CameraLog app was configured). The particle filter somewhat compensates for this by increasing the added motion noise with larger movements, but fewer samples means fewer successfully detected landmarks in general, although the slower walking speed of the blind participant also helped even things out. Despite all of this, the frame rate did not appear to be correlated with distance or time to correct convergence.

### 3.5. Performance Comparisons with Other Computer Vision-Based Localization Approaches

There are comparatively few published results of which we are aware that quantify the localization performance of other computer vision-based algorithms in indoor settings similar to ours. Moreover, because the ground truth information we use to estimate our localization error is only accurate itself to about 1 meter, it is difficult to make direct comparisons with other studies in which localization error is quantified to higher accuracy, such as centimeter- or even submillimeter-level accuracy. However, we note that the commercial smartphone-based localization app, GoodMaps Explore, which uses a 3D scanned model of the environment to enable localization, claims that “GoodMaps Explore locates where you are in a room within two inches to one meter of accuracy” (https://www.goodmaps.com/apps/explore, accessed on 17 February 2023). This is broadly consistent with the localization performance we have measured in the global localization condition. We are unaware of statistics evaluating the time (or distanced walked) required for the app to converge on a localization estimate after it is launched.

### 3.6. YOLO Performance

Landmark detection is a critical component to our approach, so we also briefly evaluated the performance of our trained object detection models. We sampled a small percentage of the frames and took a qualitative look at the bounding boxes in terms of size and position, and did a quantitative analysis of false positives, false negatives, and misclassification. Of the approximately 82,000 frames processed over all of our trials, we examined 1400 frames individually to assess the performance of our landmark detection system, randomly selecting a time step from each of 28 trials and examining the 50 frames starting from that point.

There are a few distinct ways the YOLO object detector tends to fail. First are false negatives: YOLO may completely miss an object and fail to return a bounding box on that object. Similarly, YOLO may produce false positives, returning a bounding box where there is no object. We also encounter the situation where a bounding box is returned in the correct location, but the wrong class has the highest confidence score; although this might be considered a simultaneous false negative for one class and false positive for another, this situation of misclassification tends to arise due to the architecture of YOLO. One might also consider the quality of bounding boxes for correctly identified objects: is the box in the right location, and does it tightly bound the object? We noted during development that YOLO has specific tendencies to sometimes correctly identify that an object is in the image, but displace the bounding box by an object-width or two, or to return two partially overlapping bounding boxes for the same object class.

Because the bounding box of a detection is used to estimate the position of the detected landmark with respect to the camera, we looked at the box height and box center. Without formally assessing the intersection over union (IoU) metric, we observed that in all of the examined frames the center of the bounding box was contained within the boundaries of each detected object. Qualitatively, the height of the bounding box is a little less reliable than its center location. In rare cases, the bounding box was slightly smaller than the actual object in the image, but more commonly we observed bounding boxes to be larger than the object (and for this reason we also implemented a version of our model that did not include the distance estimate in particle scoring, and although it was rather successful, we have omitted a full discussion of this experiment for brevity).

Between the different kinds of errors in landmark detection, false alarms and misclassification are much more detrimental to the localization problem than a missed detection, so we conservatively set our confidence threshold (for accepting a detection) to be 70% based on observations using our training dataset. At this value, for the sampled frames which we inspected, we found no false detections and no misclassifications (for all detections reported by YOLO, about 2% were false alarms or misclassifications, but none met the threshold of confidence). For objects we identified by inspection, only 48% were correctly identified with confidence above the threshold (with 36% being correctly identified but below threshold), less than 2% misclassified (but still below confidence threshold), and 14% were missed completely.

Although these results indicate that YOLO is prone to false negatives, the great advantage of the localization algorithm is that it is robust to errors in individual frames, especially because landmarks tend to be detected successfully in at least a few frames out of a sequence of several consecutive frames in which they are visible.

## 4. Discussion

We have designed our localization algorithm to have properties that we feel are well suited for use in a practical smartphone wayfinding app that could be used effectively by people with visual impairments. The most important benefit of the localization approach is that it is lightweight in that it requires only a comparatively small amount of input data. To create a complete localization model for a given building, the only data needed are 2D maps of each floor, along with training images for visual landmarks (just a few to several training images are needed per visual landmark) and annotations of the landmark locations and orientations on the map. In case floor plans are unavailable, it is straightforward to use ARKit or ARCore-enabled smartphone apps such as magicplan to scan the environment and generate maps (https://blog.magicplan.app/why-apples-lidar-scanner-opens-up-a-brave-new-world-of-indoor-mapping, accessed on 17 February 2023), and this capability is available even for devices without LiDAR. Once this data has been collected, a complete localization model can be generated in an automatic, offline process entailing training YOLO models; implementing this as a turnkey process is a topic for future work. Because the localization model is lightweight, it will be straightforward for users to download the appropriate model for each building of interest on their wayfinding app (perhaps chosen automatically by a GPS-indexed database, or a QR code at the building entrance or reception area).

We contrast our lightweight approach with the detailed 3D scans required by ARKit (e.g., by using persistent world mapping data (https://developer.apple.com/documentation/arkit/data_management/saving_and_loading_world_data, accessed on 17 February 2023), which we refer to as ARKit mapping, described in greater detail in Section 4.1) or other localization approaches such as Goodmaps (https://www.goodmaps.com/, accessed on 17 February 2023). We note that our approach relies heavily on VIO, which is a key feature of ARKit that provides information about relative movements in the environment, without using the ARKit mapping feature, which constructs a 3D model of the environment and performs localization by using this model. A useful property of 2D maps is that they encode the gross internal structure of a building, which changes only when the building is remodeled or renovated. Two-dimensional maps are thus likely to remain up to date longer than 3D scans, which may include incidental structures such as tables, chairs and shelves, and superficial visual features such as carpets and posters, that are frequently moved or changed. Reliance on detailed and comprehensive 3D scans may also invite privacy concerns that are largely eliminated by the use of 2D maps. Note also that the person responsible for building the localization model (such as the building manager) is free to include or exclude any visual landmarks desired, and it is straightforward to update the localization model as landmarks move or change. Although the current implementation of our algorithm assumes that the visual landmark classes are chosen to be easily distinguishable by YOLO, it should be straightforward in future work to accommodate whatever landmark classes are chosen by the building manager, whether they are easily distinguished or not. Our algorithm could be modified to consider multiple classes for each detection rather than only the most likely, and a class confusion matrix generated by YOLO on the landmark training examples can be used to accommodate ambiguous class assignments of a given visual landmark.

The design decisions of a likelihood model and particle filter clearly influence the performance of the system as a whole. We make minimal assumptions in our prior probability distribution and thus on initialization we sample uniformly in location and bearing. Consequently, despite the fact that we limit our trials to the public areas (e.g., hallways), we still populate particles in any nonwall space, such as private offices, and a different choice of prior could be used to reflect the increased probability of exploring a hallway instead of private spaces.

We observed some recurring behaviors in the particle filter that sometimes led to unsuccessful localization. A common issue in particle filters is the particle-deprivation problem [24], in which there are no particles in the vicinity of the correct state. This issue tends to arise when there are too few particles to cover the space of all states with high likelihood, and we found in our trials that this was triggered by incorrect association to recurrent landmarks. In ambiguous cases in which there were multiple instances of the same landmark (e.g., exit signs), particles in the incorrect location would be scored as high likelihood (exacerbated at times by erroneous distance estimates), and after a few rounds of unlucky resampling, there would be no particles near the correct location. Instead, we would find a singular cluster of particles tracking the wrong location (as can be seen in Figure 3c (Floor 1)), and typically this cluster would eventually hit a wall, eliminating all particles, and we would then reinitialize the particle filter. A different design for resampling might alleviate this, such as generating new particles by sampling from a measurement-based distribution, in which given a detected landmark we introduce a small number of particles in the vicinity implied by the detection.

### 4.1. Comparison between Our Approach and ARKit Mapping

In this subsection, we describe the advantages of our algorithm relative to ARKit mapping. The first advantage is that, because 2D floor plans are already commonly available for most buildings, there is no need to scan an entire building interior in 3D. Instead, the 2D floor plan must be augmented with locations of points of interest (such as offices, bathrooms, etc.) to make semantic labels available for people navigating the space using a localization algorithm, which also needs to be done by using an ARKit mapping approach. Moreover, the locations and orientations of visual landmarks must be added to the 2D floor plan, but this is only a sparse set of landmarks. Note that superficial changes to the appearance of the indoor of a building, such as new carpet or wallpaper, could necessitate a new 3D scan for the ARKit mapping approach, whereas such changes do not affect the floor plan; indeed, changes to visual landmarks, either in terms of their appearance or location, are easy to incorporate in our model and do not require changes to the floor plan itself. Finally, note that some visual landmarks, such as exit signs, are standardized and have the same visual appearance throughout a geographical region (such as California). In this case, the YOLO model can be trained on a few standard, preexisting training images of each visual landmark, rather than requiring new training images to be captured of each landmark.

Secondly, an ARKit map consists of a detailed 3D point cloud of feature points in the environment, and the maximum size that this map can represent is limited [30]. This ARKit map size limit is undocumented, but Gabriele Galimberti (personal communication, Feb. 11, 2023) has performed experiments with ARKit mapping suggesting that the maximum size of the ARKit map point cloud appears to be well under 10k feature points, corresponding roughly to a maximum surface area of several hundred square meters; attempts to “grow” an ARKit map beyond this limit results in certain feature points automatically being deleted (typically but not always the feature points that were created at the beginning of an ARKit scanning session). However, 2D floor plans are much smaller (hence our characterization of our approach as “lightweight”) and easier to scale to large environments. Although the size of a YOLO model trained to recognize a dozen classes of visual landmarks is appreciable (64 MB), the “data footprint” needed to train this model is small, because only a few sample images are needed to train each class. Overall, it is easy to acquire the data needed for our algorithm, and does not entail scanning an entire indoor environment. Because the 2D floor plan is more robust to superficial changes of an indoor environment than a 3D scan, our approach is less likely to require refreshing the entire model when these superficial changes do occur.

Thirdly, for privacy or security reasons, a building manager may be reluctant to 3D-scan certain parts of an entire environment, such as regions that are off limits to the general public (and which may contain signs or other visual landmarks that the public should not view), but it is straightforward to create a 2D floor plan that fulfills these privacy/security requirements. For instance, a building manager may wish to create a 2D floor plan that omits certain points of interest (such as a bank vault) or visual landmarks (such as a sign on a door to an office indicating the full name of the person occupying the office), to prevent unauthorized access to this information.

### 4.2. Future Work

Having evaluated our algorithm with a variety of landmark types, we feel this work demonstrates the efficacy of our lightweight approach by using multiple visual landmarks for fast localization in a mapped environment, and it also inspires further investigation. We would like to understand how these findings apply in the case of a live wayfinding app as in [4], using this work as the localization component of a guided navigation system: whether the system can effectively determine the location of a blind or visually impaired traveler as they explore the environment unguided at first toward a particular destination. It is also of interest to understand how this approach scales with a larger environment, including multiple floors of a building (perhaps utilizing the altitude information from ARKit to identify floor transitions, in addition to information from the smartphone barometer).

In this study, we represented the floor plan as a binary image in which the “wall pixels” represented impassible boundaries. However, unlike in [4] where a visibility map is created for each landmark, we did not include constraints regarding the visibility across these boundaries. Typically, one cannot see through walls, and this line-of-sight constraint has been shown to be an effective means to eliminate particles that are separated from detected landmarks by a wall (see Figure 2 as a reference). However, we recognize there are common indoor structures such as cubicles and glass walls/doors which are physical barriers that may be optically transparent (e.g., visual landmarks that are visible above a cubicle wall under certain viewing conditions) and chose not to incorporate this kind of constraint. It is possible to use a more nuanced labeling of wall pixels on the floor plan to represent physical boundaries, visual boundaries, or both.

Many newer smartphones include additional sensors, such as LiDAR, which could be easily used to detect walls. Although limited in range (on the iPhone the LiDAR maximum range is only 5 m [31]), we note that LiDAR could help with wall–floor boundary detection, which can also be used as a localization cue and incorporated into our appearance model in addition to the visual landmarks.

We found the motion estimates from ARKit’s VIO to be quite reliable in our experiments (it certainly helped that participants were instructed to avoid jerky movements with the device). Although we initially had concerns about drift (a fundamental problem in odometry), a brief investigation found no more than few centimeters of drift over routes of more than 50 m, which we felt was easily absorbed by the particle filter’s noise model. ARKit does provide a tracking status with the VIO readout to signify if it is unable to produce reliable data. We made no explicit use of this status, but future work could use this information (or our own method to detect suspiciously unnatural motion) to set the amount of noise we add to particle motion, or introduce more new particles if the measurements are deemed unreliable. We observed two cases of gross failure in the VIO output, and in one of these cases there was no successful recovery by any of the various trials over that dataset, but future versions of the system could attempt to identify the situation and adjust the algorithm (enter a recovery mode, so to speak).

We found that including distance estimates in our appearance model resulted in better performance than with azimuth estimates only; however we did observe cases where imprecise bounding boxes caused noisy distance estimates that propagated incorrect hypotheses. We would like to improve distance estimates, perhaps with a regressor [32] that directly estimates distance from a detected landmark, obviating the need for a precise segmentation (or corner localization for rectangular landmarks) to estimate pose. Potentially the pose estimation of a landmark (used to estimate distance) could be improved in some cases by using ARKit estimates of camera orientation relative to gravity and exploiting the constraint that a rectangular sign is vertical and its vertical sides are aligned to gravity.

As hypothesized, increasing the number of visual landmarks on the map decreased the time before correct localization, but there is more to investigate regarding the benefit of using additional classes of unique versus repeated landmarks (e.g., artwork and exit signs, respectively). We certainly observed more incorrect localizations when using only exit signs than with the multiple classes, but we did not establish how much of this was due to the uniqueness of the additional classes versus the fact that simply more of the map was covered by landmarks. There is certainly a tradeoff between the value gained from having a larger number of classes and the value of making the process of constructing a floor plan model easier by requiring fewer training images.

### 4.3. From Localization to Accessible Navigation

As mentioned at the end of Section 1, the motivation for developing an accurate, lightweight localization algorithm is that it can serve as the foundation for a fully accessible navigation app. Such an app can provide verbal turn-by-turn directions to the destination specified by the user, based on the location estimates from the localization algorithm. We implemented a prototype navigation app in [22] by using an earlier version of our localization algorithm [4] implemented on an iPhone 8. The app assumes that the user knows which building floor they are located on, and allows them to choose a destination from a pull-down menu. The user begins walking with the camera pointed forward. No navigation guidance is offered by the app until the localization algorithm has converged to an estimate. Once the location estimate is available, the app offers verbal directions such as “turn right” a short distance before the direction is to be executed. We note that this navigation app assumes that the layout of the walkable portion of the floor plan is dominated by corridors—in other words, that this portion is well represented by a graph, with nodes on the graph denoting points of interest and “control points” (locations where the user either must turn or has multiple turn options), and graph edges connecting nodes representing corridors. Future research will explore effective user interface options in open spaces such as airports, where walking paths are much less constrained than in corridor-dominated environments, so that simple directions such as “turn right” may be ambiguous.

## 5. Conclusions

In this paper, we describe a novel localization algorithm that combines visual landmark recognition, visual–inertial odometry (VIO) and spatial constraints imposed by a 2D floor plan. Building on our previous work [4], our localization algorithm now includes the ability to recognize multiple classes of visual landmarks and a more powerful appearance model that incorporates more information about how the appearance of a landmark in an image constrains the user’s location. Areas of opportunity for improvement include incorporation of visibility of landmarks (e.g., which may be occluded by intervening walls), use of LiDAR, and of course the implementation of the algorithm as an accessible smartphone navigation app. We assess the performance of our algorithm in terms of the time (and distance) required for the algorithm to converge to an accurate localization estimate and localization error, after being initialized with minimal information (such as the fact that the user is in an unknown location on the map). Experimental data collected by both sighted and blind participants shows that convergence time (and distance) decreases with the inclusion of more classes of visual landmarks. Once the localization algorithm has converged, however, the use of additional visual landmarks has little impact on the localization error. Our algorithm is fully accessible to BVI users, and our past work demonstrates that it will be feasible to port our algorithm to a smartphone for real-time implementation. This implementation will serve as the foundation for a fully accessible navigation app that issues real-time turn-by-turn directions to the user.

## Figures and Tables

**Figure 1 sensors-23-02701-f001:**
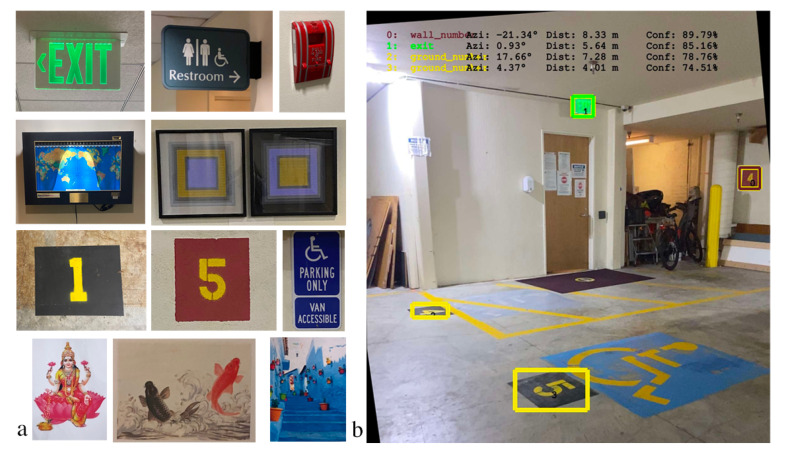
In (**a**) on the left of the figure, we show examples of 11 classes on which the YOLO object-detection network was trained. The top row shows examples of landmarks that are part of the building, the second row shows wall decorations that we found as already in the environment, the third row is informational signs in the garage environment, and the bottom row shows sample artwork that we printed ourselves to post as distinctive landmarks. The right side (**b**) visualizes the object-detection process. The image is first rotated so that it is upright with respect to gravity (using the VIO data), then passed to the YOLO object detector. By using the height of the bounding box and the horizontal position of its center, we estimate the distance and direction of detected landmarks.

**Figure 2 sensors-23-02701-f002:**
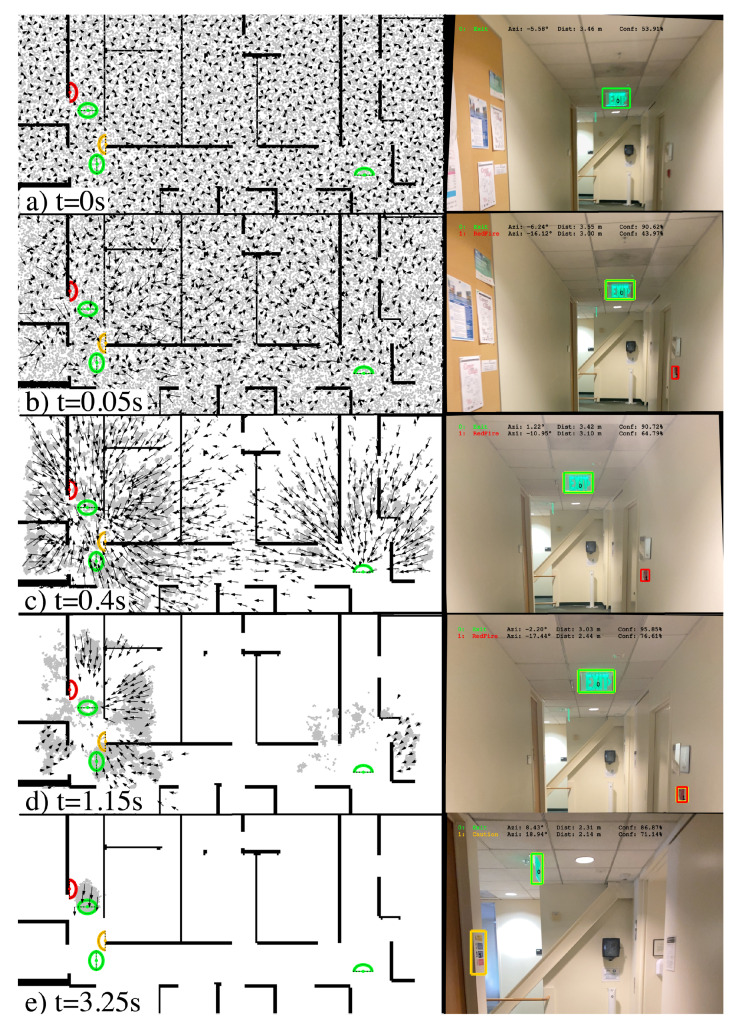
This figure shows how the particle filter accumulates localization evidence over time. It begins at t=0 s. with maximum pose uncertainty and proceeds to four subsequent moments in time, the last of which (t=3.25 s.) is associated with high pose certainty. See text in Section 2.2.5 for details.

**Figure 3 sensors-23-02701-f003:**
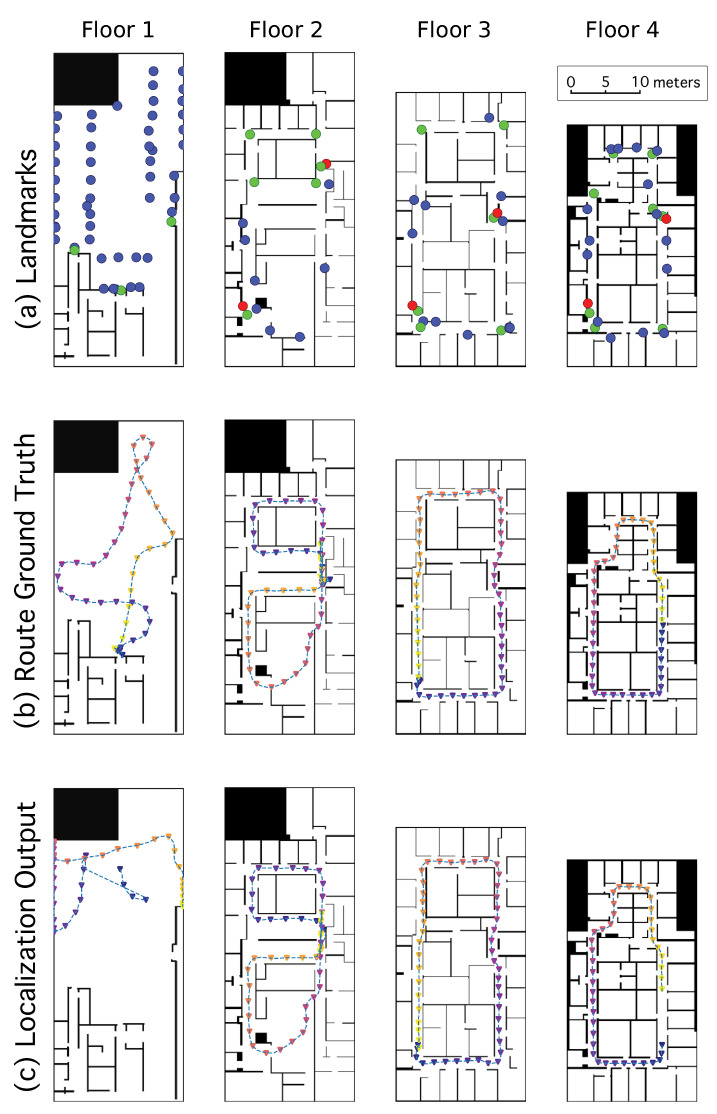
Each column shows a different map used in our trials (floors 1 through 4 are displayed left to right), the first being an open environment and the rest having corridors. In row (**a**), we present the locations of visual landmarks: exit signs and fire alarms are marked in green and red, respectively, with all other classes marked with blue. Row (**b**) shows a sample route from each map, markers show the ground truth position (sampled every two seconds) with a color gradient from blue to yellow to indicate the passage of time. We show example output trajectories from our algorithm in row (**c**). Floor 1 demonstrates a failure case in which the algorithm converges, but in the wrong location. Floor 4 shows a case that is successful after some time before convergence, and Floors 2 and 3 show successful cases of quickly converging on the correct location.

**Figure 4 sensors-23-02701-f004:**
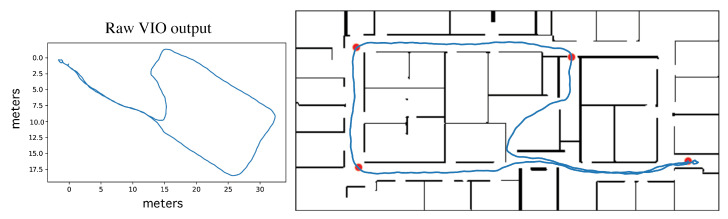
The **left** plots the *X*, *Z* coordinates of the device as reported by VIO in ARKit’s gravity-oriented coordinate system (the *Y* coordinate is omitted because this is aligned to the vertical direction defined by gravity). The **right** shows a mapping of these coordinates overlaid on a floor plan map. To determine the mapping (an affine transform that includes inverting one axis, rotation, scaling, and translation), we observed the location of the participant in the building on a recorded video of the data collection and then marked positions on the map, which would correspond to the reported VIO coordinate (these positions are shown by red dots). Note that the (X,Z) coordinate system used by VIO is inverted with respect to the (x,y) coordinate system used in the map.

**Figure 5 sensors-23-02701-f005:**
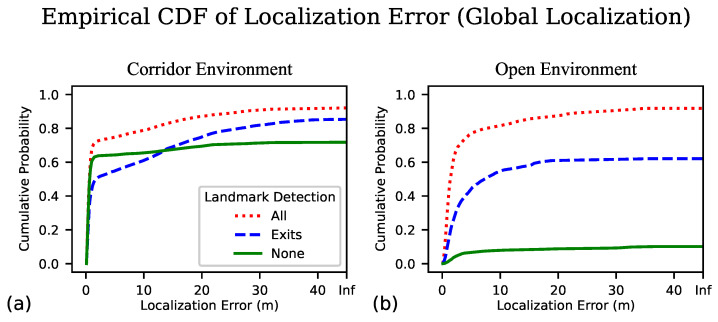
The empirical cumulative distribution of localization error, compared against different landmark detection modes: all available landmarks, exit signs only, and no detection. (**a**) On the left is in the hallway corridor environment and (**b**) on the right is in an open space.

**Figure 6 sensors-23-02701-f006:**
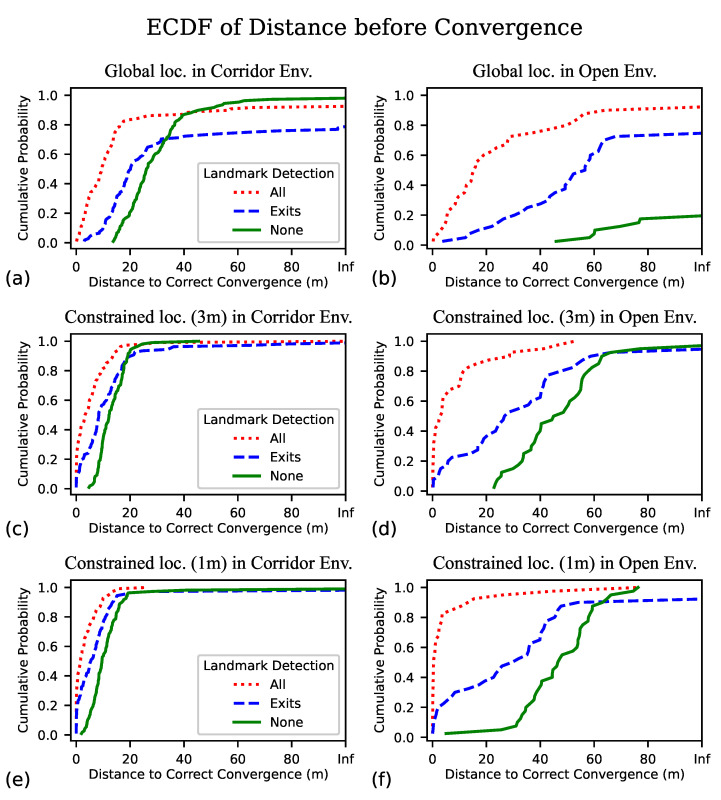
The empirical cumulative distribution of the distance traveled before the algorithm correctly determines the user’s location, compared against different sign-detection modes: all available signs, exit signs only, and no detection. On the left (**a**,**c**,**e**) is the hallway environment and on the right (**b**,**d**,**f**) is an open space. The first row shows results for the global localization problem whereas the others below show constrained localization (at 1 and 3 m).

**Figure 7 sensors-23-02701-f007:**
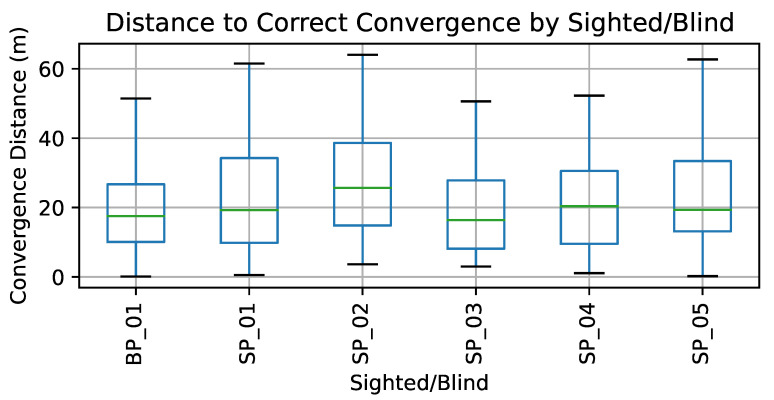
We compare the distance to correct convergence for each of the participants. We note that there is no obvious difference in the algorithm performance for data collected by blind participant (BP) or sighted participants (SP).

**Figure 8 sensors-23-02701-f008:**
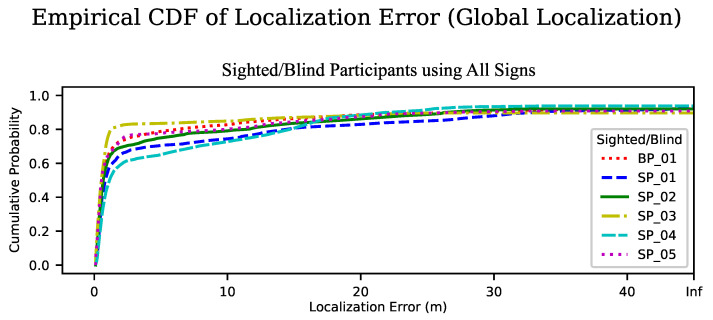
We compare the localization error for each of the participants via the ECDF, specifically the case of all signs. Although there is some variation between participants, the blind participant scores in line with the others.

**Table 1 sensors-23-02701-t001:** Table of results over the key metrics of median time/distance to correct convergence, median localization error and total rate of correct convergence. For the global localization condition, we divide the data by area type (either open or corridor environments) and then by what type of landmark detection was employed (all classes of landmarks, exit signs only, or no landmark detection).

Area Type	Landmark	Convergence	Convergence	Localization	Convergence
	Detection	Distance (m)	Time (s)	Error (m)	Rate (%)
**Corridors**	**All**	8.9	14.7	0.4	91.7
	**Exits**	17.9	30.1	0.6	78.7
	**None**	26.0	46.2	0.4	97.2
**Open**	**All**	14.7	24.9	1.2	90.0
	**Exits**	49.2	60.4	2.0	72.5
	**None**	60.3	89.0	3.8	17.5

**Table 2 sensors-23-02701-t002:** Metrics for the constrained localization case, in which we assume the starting position is known within 1 or 3 m of ground truth location (bearing still unknown), compared between both open and corridor environments and type of landmark detection that was employed (all classes of landmarks, exit signs only, or no landmark detection).

Area Type	Landmark	Convergence	Convergence	Localization	Convergence
	Detection	Distance (m)	Time (s)	Error (m)	Rate (%)
Initial Radius 1 m					
**Corridors**	**All**	1.6	3.2	0.4	100.0
	**Exits**	5.2	8.5	0.5	97.2
	**None**	9.5	16.2	0.4	98.1
**Open**	**All**	0.4	1.1	0.9	100.0
	**Exits**	25.2	32.3	1.4	90.0
	**None**	46.9	71.4	1.2	100.0
Initial Radius 3 m					
**Corridors**	**All**	3.2	6.5	0.4	99.1
	**Exits**	8.3	15.2	0.5	98.1
	**None**	12.3	21.3	0.4	100.0
**Open**	**All**	3.1	4.4	1.0	100.0
	**Exits**	26.2	33.2	1.4	92.5
	**None**	44.6	66.8	1.4	95.0

## Data Availability

The data presented in this study are openly available in the Inter-university Consortium for Political and Social Research (ICPSR) repository at https://doi.org/10.3886/E183714V1 (accessed on 17 February 2023). This includes the maps and landmark annotations, objection detection models as Core ML files, text readouts of ARKit’s VIO data and accompanying images, and Python source code to parse the data, run the particle filter, and compute the reported metrics.

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
