# Peer review of "A Lightweight Approach to Localization for Blind and Visually Impaired Travelers"

_sensors, 2023, doi:10.3390/s23052701_

Round 1

Reviewer 1 Report

The authors present an algorithm based on computer vision and inertial sensing, requiring only a 2D floorplan of the environment annotated with the locations of visual landmarks and points of interest. The work is interesting as it tackles the practical problem of wayfinding for visually impaired people. Some important points should be clarified so readers can improve their understanding about the text, as listed as follows.

The title of the paper uses the term "lightweight" but no information regarding performance of the proposed algorithm is provided. Authors mention the performance of their previous work (10fps on an iphone 8) but do not provide data regarding performance of the new version.

How does the proposed algorithm deal with the problem of having the same landmark (exit sign) in more than one location in the same floor? Or how does the proposed algorithm work when both sides of the exit sign are the same? How are the probabilities affected because of that fact? Is that a limitation of the proposed approach?

It is not clear how BVI users may access the algorithm. Is there already a high level app running on a cell phone? The authors mention as future work to create such application. I was able to see how it should work in authors' previous paper, but some description about it should be added to the current paper.

It would be nice to have some comparison with related work. For example, what are the gains of the proposed algorithm in comparison to simply using the ARKit mapping? Since the landmarks have to be captured and located on the 2D map, it is a similar work having to walk inside the environment to construct the map to be used by ARKit. Please highlight the advantages of the proposed approach in comparison with other works. What is the comparison to previous work of the same authors, for example?

The authors should avoid the excessive use of loose sentences inside parentheses. Please remove those sentences from the text or add them by removing the parentheses symbols. Some occurences of those are listed as follows.

(The focus of this paper is on wayfinding rather than the development of mobility aids to warn travelers about nearby hazards such as obstacles and drop-offs [3].)

(In future work we hope to augment our localization approach with inertial sensing to lessen its dependence on video imagery, which can be challenging for the user to acquire while walking. While inertial sensing is already integral to the Visual-Inertial Odometry approach we use in our localization algorithm, the use of the IMU to interpret gait movements would allow our algorithm to function even in the absence of video data, e.g., when the smartphone is carried in the pocket or purse.)

(However, it is straightforward to update the localization model to add or change the visual landmarks.)

(It is closely related to the yaw estimated by iOS ARKit, as we will describe later; note that the bearing is a 2D concept, and it is undefined if the camera line of sight is perfectly vertical.)

(There may even be inaccuracies in the map itself, which adds additional uncertainty.)

(This 3D pose shouldn’t be confused with the full 6D pose associated with VIO.)

(The floor plan is represented as a binary image, with white pixels corresponding to open space and black pixels corresponding to walls and other barriers, so detecting such intersections amounts to a simple ray-tracing calculation.)

(See Appendix A for details.)

(We use the GravityWorld Alignment option in ARKit11, which uses a coordinate system in which Y is aligned with gravity and the X, Z coordinates are arbitrarily aligned to the horizontal plane rather than to the 2D map coordinates (x, y).)

(See [28] for a tutorial on particle filters.)

(The rationale for this approach is that missing location estimates could 579

be replaced by a default location estimate, such as the centroid of the map, and the resulting 580

localization error would be guaranteed to be no worse than the “Inf” value.)

(Implementing this as a turnkey process is a topic for future work.)

(Our algorithm could be modified to consider multiple classes for each detection rather than only the most likely, and a class confusion matrix generated by YOLO on the landmark training examples can be used to accommodate ambiguous class assignments of a given visual landmark.)

More general comments and writing errors are listed as follows.

"the user to the vicinity of their" -> "users to the vicinity of their"

"landmarks need be photographed" -> "landmarks need to be photographed""the user to the vicinity of their" -> "users to the vicinity of their"

"landmarks need be photographed" -> "landmarks need to be photographed"

"to to BVI users" -> "to BVI users"

"over time to arrive at a location estimate (x, y) relative to a 2D floor plan (which we refer to as the “map”) that is tracked over time" -> please rewrite

"the participant was told" -> "the participants were told"

It seems that the paths from both images (Figure 4) are mirrored.

"We are more concerned with the incidence of catastrophic errors – highly consequential errors that could send the traveler down the wrong corridor – than with the incidence of small ones, and our performance assessment reflects this priority." -> what do you mean by "small error"? What is the magnitude of such value?

"than simply omit" -> "than simply omitting"

"So despite" -> "So, despite"

"we still" -> ", we still"

"we note that" -> ", we note that"

"In case floor plans are unavailable, it is straightforward to use LiDAR-enabled smartphone apps such as magicplan to scan the environment and generate maps16." -> this could be also done using ARKit or ARCore, with no LiDAR at all.

Reviewer 2 Report

In the introduction section briefly describe the advantages and disadvantages of the technologies have been developed for way finding applications.

1. In the conclusions section, describe specifically 3 advantages of the technology developed with respect to previous technologies and 3 areas of opportunity for improvement. 2. Try to improve the resolution and size of the figures. 3. Synthesize a little more the description of the mathematical foundations.

Reviewer 3 Report

1. The English presentation of the contents need revision (see examples below) for professional academic purpose; please proofread the entire paper.

"enforcing the fact that the apparent height (the difference between the bottom and top pixel row coordinates of the landmark’s bounding box in the image) increases as the landmark approaches the camera. Because of variations in the bounding box size returned by YOLO (discussed in 3.5), and because of situations in which the bounding box is poorly aligned to the borders of the landmark (e.g., the yellow “5” sign shown in the bottom of Figure 1b), the distance estimate is noisy. As a result, the distance estimate is not weighted strongly in the model."

"This estimate tended to be less noisy than that of the distance, so it is weighted more heavily in the model."

2. It is unclear why AdaBoost is changed to YOLO; please incorporate the rationale or cite the section explaining that.

"(The main difference with respect to the current algorithm is that the old implementation used an AdaBoost-based Exit sign recognition approach, whereas the new localization algorithm uses YOLO to recognize multiple classes of visual landmarks.)"

3. The authors put a link to [15], but it links to another paper that does not have any of the authors of this journal paper under review.

"Our Python source code, along with maps, trained Core ML models, ARKit’s VIO data and accompanying images have been made publicly available15".

"15. Khan, D.; Ullah, S.; Nabi, S. A generic approach toward indoor navigation and pathfinding with robust marker tracking. Remote Sensing 2019, 11, 3052."

4. With six participants, the data collected is not comprehensive to inform the audiences of the performance of the proposed localization algorithm. Suggest including the comparison results with other existing localization algorithms used to detect similar areas and landmarks in Table 1 and 2. 

5. The Abstract should indicate the performance improvement, e.g., in %. The authors claimed this without further mention.

"In this work, we improve upon the existing algorithm so as to incorporate recognition of multiple classes of visual landmarks to facilitate effective localization, and demonstrate empirically how localization performance improves as the number of these classes increases."
